# Appraising and Handling COVID-19 Information: A Qualitative Study

**DOI:** 10.3390/ijerph181910382

**Published:** 2021-10-02

**Authors:** Morhaf Al Achkar, Matthew J. Thompson, Diem Nguyen, Theresa J. Hoeft

**Affiliations:** 1Department of Family Medicine, University of Washington, 331 NE Thornton Place, Seattle, WA 98125, USA; mjt@uw.edu; 2Department of Global Health, University of Washington, Seattle, WA 98125, USA; diemnguyen.2011@gmail.com; 3Department of Psychiatry and Behavioral Sciences, University of Washington, Seattle, WA 98125, USA; thoeft@uw.edu

**Keywords:** COVID-19, risk communication, health information

## Abstract

Background. The coronavirus pandemic brought vast quantities of new information to the public for rapid consumption. This study explored how people most impacted by the pandemic have judged and perceived the quality of information regarding COVID-19 and regulated the information flow. Methods. This was a qualitative study of semi-structured interviews developed as a pragmatic study targeting several groups most impacted by the pandemic. Participants were identified through convenience, purposive, and snowball sampling methods. They were interviewed by phone or video conference. Results. Twenty-five participants were interviewed between 6 April 2020 and 1 May 2020. In terms of verifying information and judging its quality, people judged information by the source. People compared information across sources and attempted to verify the quality. Most felt self-assured about their capacity to judge information. Regarding the quality of information, many participants felt the information was skewed or inaccurate. Contradictory information was confusing, especially with a strong suspicion of ulterior motives of information sources impacting trust in the provided information. Yet, some recognized the iterative process of healthcare-related information. In terms of regulating information flow, many participants perceived flooding with information. To counter information overload, some became selective with types of information input. Many developed the habit of taking breaks periodically. Conclusion. Improving risk communication in a pandemic is of paramount importance. Organizations working in public health must develop ways to regulate information flow in collaboration with trusted community partners. Individuals also must develop strategies to improve information management.

## 1. Introduction

The coronavirus pandemic brought vast quantities of new information to the public for rapid consumption. The disease has led to over 500,000 deaths in the United States, where deaths per capita have surpassed many other countries [1,2]. Throughout the pandemic, the rapid and ever-changing nature of information regarding COVID-19 has been overwhelming. Individuals and systems had to keep abreast of its spread, hospitalizations, mortality rates, “hot spots”, necessary precautions, and adaptations to social activities and work [3,4]. Information consumption by individuals depends in part on personal circumstances and health risks [5]. Additionally, individual- and system-level health literacy (in the sense of knowledge and competency to access, appraise, and apply information to health decisions) plays a major role in choices [6,7]. Systems also recognized the need to quickly adapt to and rapidly disseminate changing information [3]. As the volume of information rose, systems had to employ new knowledge in real time, which posed challenges for individuals on many levels [8]. This has been even more burdensome because of the questionable quality, validity and understandability of some information made public over this time [9,10].

Because pandemics and natural disasters are periodically expected crises, recommendations for delivering messages to the public via trusted channels have been developed as part of risk communication strategies [11]. According to Abrams and Greenhawt [5], risk communication is defined as the “exchange of real-time information, advice, and opinions between experts and people facing threats to their health, economic, or social well-being.” Two-way or multi-directional communication of risk has been highlighted as critical in response to COVID-19 [12], focusing on rumor management, improved communication across agencies, and consistent messaging from the health care and private sectors [13]. Following Hurricane Katrina, for example, reports questioned the media’s role in contributing to rumors [14]. Beyond mass media, the COVID-19 pandemic has been marked as an “infodemic” resulting from widespread distribution of unvetted information. However, little is known about how the public and health care workers have sought and managed information during the pandemic.

In order to better inform future messaging efforts in this rapidly changing environment, we explored how the people most impacted by the pandemic have (1) judged and perceived the quality of information regarding COVID-19 and (2) regulated the information flow. We anticipate that the results of this study will provide a deeper understanding of how these processes can inform a multi-stepped approach to reaching a broader population, especially hard to reach groups, through trusted channels with clear, persuasive and culturally relevant messaging.

## 2. Methods

### 2.1. Study Design

This was a qualitative study of semi-structured interviews developed as a pragmatic study targeting several groups most impacted by the pandemic. We interviewed a diverse sample of participants in the United States to learn about their experiences with COVID-19, including their management of information, the pandemic’s impacts, and their unmet needs. Participant groups included health care workers, people more vulnerable due to underlying health conditions such as cancers, marginalized minorities and people of color, workers impacted by the pandemic lock-down, and others. 

### 2.2. Study Population

Participants met broad inclusion criteria: (1) older than 18 years; (2) psychologically and physically well enough to participate; and (3) English speaking. We identified participants through convenience, purposive, and snowball sampling methods. While we had broad inclusion criteria, reflecting the widespread impact of the pandemic, we attempted to recruit people who could speak of particular group experiences. In order to sample diverse perspectives, we used a variety of recruitment strategies, including reaching out to active users of social media (LinkedIn, Twitter, and Facebook) with opinions on the subject, listservs of health care providers, personally known community advocates, gatekeepers to patient communities, participants who could identify other potential participants, and other strategies.

### 2.3. Study Procedures

Participants were interviewed by phone or video conference. Verbal consent was obtained at the start of the interviews. We asked participants about how they obtained and managed information at the early phase of the pandemic. Each interview was audio recorded and transcribed. The interview guide is included as an Appendix A. Participants were reimbursed $25 for the interview.

### 2.4. Analysis

We used NVivo 11 (QSR International, Burlington, MA, USA) to organize the qualitative data and conduct the analysis, which was conducted alongside the data collection. We used inductive and deductive thematic analyses. We specifically looked at data quality, quantity, and sources of information. Low-level codes were ascribed to the text as outlined by Carspecken [15], and the coded text was extracted and further explored to uncover themes and subthemes. We organized the findings to highlight information sources, quality and quantity of information, and how people managed information flow. Two authors (MA, TJH) engaged in peer debriefing to review aspects of the work, including coding, theme development, and findings.

## 3. Results

We interviewed 25 participants between 6 April 2020 and 1 May 2020. Table 1 includes participant characteristics. Sources of information and types of information sought are included in Table A1 and Table A2 in the Appendix A. 

In terms of verifying information and judging its quality, four themes emerged. Table 2 presents supportive quotes.

**People judged information by the source.** Everyone had trusted sources and standard places from which they gleaned information. People trusted medical experts, especially those experts with years of experience. Some also trusted their personal doctors, especially those who they perceived would listen. Many people trusted the Center of Disease Control (CDC), although some did not trust anything from the government. Some recognized that people’s political views influenced their choice of information sources.**People compared information to information**. To make a decision, people sought to compare different types/sources of information, for instance, comparing what they read on social media with what was presented on TV. Some looked at both liberal and conservative publications or looked at a certain number of resources, for example, ten resources to see if, say, four agreed. For others, the principle was “the truth is in the middle,” and they sought a balance.**People attempted to verify the information.** When presented with information, some individuals wanted to verify it through their own research, including PubMed searches of original studies. They found trustworthy studies or sites, and many looked for statistics to study the numbers. They appreciated clear methodologies and sought what they considered unbiased work.**Most felt self-assured about their capacity to judge information.** While not all had medical qualifications or received training in public health, everyone processed information and made decisions based on their appraisal. Respondents were proud of their skepticism; many felt confident and qualified in consuming information. Some recognized that the public did not understand the scientific process of generating knowledge and the iterative back and forth nature of research. Many individuals’ perception was that people make decisions based primarily on their proximity to the pandemic or the personal impacts of the virus in their smaller circles.

Regarding the quality of information, five themes emerged. Table 3 presents supportive quotes.

**Skewed or inaccurate information and misinformation were abundant.** Most participants were concerned about the media spreading misinformation, especially from some officials who appeared to be cavalier about the pandemic. Many found the information to be opinion based rather than factual. Some thought case numbers were inflated, whether intentionally or due to using different methodologies. Still, some found the information provided to be reliable.**Contradictory information was confusing.** Participants complained about receiving mixed signals from different media outlets and information sources. They found it confusing when each outlet reported different information, making it hard to trust the unclear and messy recommendations.**Ulterior motives of information sources.** Many worried that news focused on using information to attract viewers, while others considered that the underlying agendas of politicians or businesspeople may be influencing the information provided.**Many did not trust the information provided.** With skewed, contradictory, and unclear motives, people had a difficult time trusting the information they were provided, and many thought they were not told everything or that the pandemic was worse or better than the information they were seeing.**Some recognized the iterative process of health care-related information.** Some recognized that entities such as the CDC and state governors produced concise and clear recommendations or that the information evolved over time. Others acknowledged that information is changeable by nature and that many health-related groups or organizations were discovering and applying new information in real time.

In terms of people needing to regulate information flow, four themes emerged. Table 4 presents supportive quotes.

**Flooding with information.** At first, people dealt with an abundance of information that they found overwhelming. Many sought to read everything throughout the day and felt they could not stop. It was hard not to keep checking, especially with information changing rapidly. They felt that multiple groups, entities, and individuals had something to say. Within this abundance of information, many perceived that quality evidence was scarce.**Being selective with types of information input.** Over time, some people started to be more selective about where they obtained information. For example, limiting it to trusted resources such as university emails for those working within universities. Some began avoiding sensational and attention-grabbing information sources; instead of following every thread, they focused on learning what was useful in terms of what they could control.**Regulating the amount of information.** In addition to greater selectivity with types of information, individuals started filtering the volume of information presented to them and scrolling or scanning headlines instead of reading everything. Some returned to regulating their news as they did pre-COVID-19. A few realized they saw the same information repeated over and over. They determined how much to read and avoided constantly listening to the news.**Taking breaks.** Many took breaks from the information, turning off the TV and avoiding the news. They also avoided social media.

## 4. Discussion

Our study is the first to use interviews and qualitatively explore the perceptions of a broad range of groups, including health care workers, Black communities, and cancer groups, regarding pandemic-related information management and flow. Our study provided an opportunity for a timely assessment of public and health care worker perceptions and has numerous practical implications.

People used various sources to obtain information and moved from one source to another to verify reliability. Our study was consistent with the literature regarding information sources during the pandemic [16,17,18,19]. It also provides a more expansive view of information sources compared to other studies with narrower foci (e.g., types of social media) [17,20,21]. Previous studies of online COVID-related information raised alarms. A significant proportion of what was provided on Twitter, YouTube, and other websites was considered misinformation or misleading, unverifiable, and low-quality information, or it was written in language that was more complex than readability standards [10,16,20,22,23,24].

Some participants confirmed expert opinions that what appeared to be confusing information was part of the natural, iterative process of evolution of knowledge. While recognizing the need for fast-paced dissemination of new information, voices in the information community have called for maintaining quality and following ethical standards, including exerting the greatest efforts to ensure the validity and methodological rigor of what was published [4,9]. Not only was the quality of information problematic, but our study also highlighted concerns of information flooding [18,22].

The health care system had to adapt quickly and through iterative processes that required extensive daily communication as information changed rapidly [3]. Our work is also consistent with concerns that the quantity exceeded an individual’s capacity to grasp and conceptualize [5,16], where multiple groups put out public guidance and clinical guidelines. According to Wang, information overload became a problem, and worries about people’s ability to respond were valid. Kearley warned of “alert fatigue,” and our participants complained of feeling burdened by the massive quantity of daily emails [16]. Possible solutions include using “command centers” to enable a hierarchical and regulated flow of information, organizations issuing joint recommendations, and developing practice algorithms into one-page concise references [4].

Our study shows that individuals felt the burden of information overload when they had to spend many hours every day seeking or receiving updates. Yao correlated hours spent per day receiving information with psychological distress [25]. The association between information seeking, worries, and preventive behavior is quite complex, and the causality is multi-directional [26,27]. Our research also suggests that people were porous to new information at first. However, driven by the perception of an information burden, some adapted by becoming selective about time and content, and others took regular breaks to detach and recharge. The adaptive patterns exhibited by some of our participants are consistent with self-regulation and developed self-efficacy.

The findings of our work suggest three main practical recommendations to improve risk communication in a pandemic. First, state, county and local public health offices should partner closely with community agencies to create brief messages that are culturally relevant and at the appropriate literacy level. Second, consideration must be taken to develop ways to regulate information flow by communicating across agencies and even within departments of the same organization. Third, people should not only be encouraged to take breaks from media and information about COVID-19, but should also be offered comprehensive sources of simplified COVID-19 information from diverse trusted sources that they can reference.

Our study has many strengths. We interviewed a range of health care providers and individuals in the general public. Their contrasting experiences provided an opportunity to demonstrate a spectrum of patterns that made it possible to understand the diversity of information flow experiences. Our participants were quite diverse in terms of race, job, political opinions, and educational attainment. This diverse sample supports the transferability of our findings and their relevance to a broader public. By triangulating perspectives of healthcare workers with those of the public, we showed the commonalities and depth of the lived experience.

Nonetheless, this work is not without limitations. The heterogeneity of the sample selection strategy led to a diverse group but may have omitted individuals with relevant experiences, including professionals in other areas, such as public transport, hypermarket workers, among others. Clearly, the 25 participants, while large enough for a qualitative study, may not have included the experiences of all sectors in a pandemic. Future work will explore the experiences of other marginalized communities and individuals that were particularly impacted by the pandemic, such as Latinos, limited English language speakers, people with low technology literacy and access, the homeless, and gig workers. Quantitative approaches may be needed to compare the experiences of different groups including contrasting the experiences of healthcare workers to other sector workers and to the public. Second, our study looks at the period of 1–2 months after the pandemic started in the United States. People’s experiences were already changing, so our study may have been more akin to a snapshot and may have not captured everyone’s full experience. Because the pandemic hit different parts of the country at different times, it is imperative to conduct follow-up interviews exploring how people’s positions and experiences are changing over time.

## Figures and Tables

**Table 1 ijerph-18-10382-t001:** Participants’ characteristics.

ID	Date of Interview	Age Range	Gender	Race	Work
1001	6 April 2020	30–40	Male	White	Family doctor
1002	7 April 2020	40–50	Female	White	IT and gig economy worker
1003	7 April 2020	40–50	Male	White	Thoracic surgeon
1004	8 April 2020	40–50	Male	Black	Retired football player
1005	9 April 2020	40–50	Male	White	Thoracic surgeon
1006	8 April 2020	50–60	Female	Black	Disability office associate director
1007	9 April 2020	40–50	Female	Black	Family doctor
1008	10 April 2020	40–50	Female	Black	Hair stylist
1009	13 April 2020	60–70	Female	Black	Retired ob-gyn doctor
1010	16 April 2020	30–40	Female	White	First-year resident
1011	16 April 2020	50–60	Female	White	Family doctor
1012	16 April 2020	30–40	Male	Black	Fellow, emergency medicine
1013	16 April 2020	20–30	Female	Black	PhD student
1014	26 April 2020	40–50	Female	White	Substitute teacher
1015	26 April 2020	20–30	Female	Asian	Intern, family medicine
1016	26 April 2020	40–50	Female	White	Stay at home mom
1017	26 April 2020	40–50	Female	White	Uber driver
1018	27 April 2020	30–40	Female	White	Stay at home mom
1019	20 April 2020	30–40	Male	White	Family doctor
1020	28 April 2020	40–50	Female	White	Volunteer, cancer research
1021	28 April 2020	30–40	Female	White	Nurse
1022	28 April 2020	50–60	Female	White	Pharmaceutical contract specialist
1023	28 April 2020	30–40	Female	White	Stay at home mom
1024	29 April 2020	60–70	Female	White	Hospice social worker
1025	1 May 2020	40–50	Female	White	Stay at home mom

**Table 2 ijerph-18-10382-t002:** Themes and quotes related to verifying information and judging its quality.

**People judge information by the source**
I see where the information is coming from. If it’s coming from just an old high-school classmate, I don’t put too much truth on it. If it’s actually on the news, I’ll listen to that a little bit more because the source is there. (1018)
Unless I’m reading something from the CDC or the World Health Organization, I’m not quite certain if I believe what I’m hearing. (1006)
I had an appointment with my Rheumatologist and he was like, “you need to be aware of this and take it seriously because this medication is suppressing your immune system, and you’re more susceptible.” Then, that’s kind of opened my eyes that I have to protect myself. (1008)
**People compared information to information**
What I’m hoping for is if I look at ten sources of information, if 3 or 4 them line up then maybe that is the information to trust. (1002)
I always try to weigh those different sources and I usually land somewhere in the middle because it seems like that’s a little more reasonable. (1020)
I do tend to balance both of my sources. I look at more liberal publications, then I’ll also look at more conserve sites, just to see what both sides are saying about the situation to try to balance out what I’m hearing from each. (1003)
**People attempted to verify the information**
A lot of times, I’m looking for research, for more medical journals. (1007)
It’s hard to find things that aren’t biased in some way but just sourcing and more stats, and what has some methodology to them. (1016)
I go through what judgments are made in the various accounts that I hear or read. If there’s something that feels kind of off, I’m going to do more research on that to verify. So I usually do a little fact-checking and my research had on and see what types of bias coming into the picture. (1013)
**Most felt self-assured about their capacity to judge information**
I’m not a black-and-white person. I’m more of an analytical person that’s trying to just get information, because those are still so many unknowns. (1016)
Unfortunately, I think a lot of people don’t source their information where they’re getting it from as far as evidence-based quality. They just go for what’s convenient and easy. They’re not going for the most healthy, nutritious thing but the quick, easy junk food information. (1005)
My impression is, if it doesn’t affect their immediate family that they don’t think it’s occurred. But it has already affected me, my immediate family and two people that I know that have actually tested positive for it. One person was my father’s relative and he died. (1022)

**Table 3 ijerph-18-10382-t003:** Themes and quotes related to the quality of information.

**Skewed or inaccurate information and misinformation were abundant.**
Sometimes the information presented to the public, especially in terms of either treatment or medication, is skewed. They are presented as being a bit more hopeful than what we’re actually seeing in the hospital from patient experience. (1015)
My impression is, the information might be a little overboard. They always say, “Oh, so many people died.” Because here in Hawaii, we’ve had 14 deaths. (1017)
When I first heard about it on the news, I thought it was the news media just trying to blow it up, I was going to go on vacation for my birthday. I was like, “Oh, whatever there’s the flu and stuff is really not that big.” I ended up going ahead and go to Miami [from Seattle]. (1008)
I would say the information I get from my company and the information I get from the Governor and the State of Connecticut are all clear, concise, and detailed. (1022)
**Contradictory information was confusing**
It seems like there’s a whole different outlets reporting different things, and so it’s a little confusing to me. I don’t know what’s really the real story going on? (1018)
I’m skeptical a lot on the ways. That COVID information comes from a government or from our governors is messy and not clear. So, is it an environmental? Or is it airborne? Or what are the symptoms? They keep changing. (1006)
There is the anxiety created by not knowing what information to trust. (1002)
**Ulterior motives of information sources**
I think the administration was confused at the beginning, plus they have their own agenda. (1009)
We had a variety of sources and some seems a little bit more interested in selling the fear factors and not as much of the actual data. (1025)
News reporting tries to generate the most number of people to tune in. People start to align themselves with what they’re comfortable with. So they’ll start watching that news feed because you’re getting more of what you agree with versus a bigger global picture. (1025)
**Many did not trust the information provided.**
I think the numbers might be a little inflated maybe because they’re not really telling us the difference between those who died and were healthy and those already sick. (1017)
When first we heard of the drugs in the news, the combo, Plaquenil and Azithro, and even Remdesivir it was kind of portrayed as a miracle treatment that’s going to cure the virus that was more like what was presented on the news and in the media, versus in the hospital, we were treating it as more of just a trial and error. (1015)
**Some recognized the iterative process of learning in health systems**
Everything’s just been changing as more information comes out. We started a few weeks back giving every patient Azithromycin and Plaquenil. Then we changed to giving only Plaquenil and then of late we’re not even giving Plaquenil as more information is coming out that it may or may not be improving patient’s condition. (1015)
I think because it’s a lot of opinion-based and not a lot of good evidence out there. It’s still so early. It’s been interesting to see the case reports and small series and to try to tune that into what we’re seeing here. (1003)
We do that a lot with cancer research. We have a standard way of thinking that may last for years or decades even and then all of a sudden someone will make a discovery and go “oh, oh that’s wrong”. Forget all the things we told you before but it’s more subtle normally and it’s taking place over longer periods of time and so it doesn’t seem quite as shocking as it does right now with the work being done around COVID. From one day to the next, the story will change completely. I think that just has to do with how much data is being shared and how many people are working on the problem. (1020)

**Table 4 ijerph-18-10382-t004:** Themes and quotes related to the regulation of information flow.

**Flooding with information**
I just started reading every day, all day long, anything I could get my hands on, whether that would be the news, newspapers, articles, professional articles. (1007)
As soon as the Johns Hopkins site was up, I was tracking it every day. And then lots of stuff in The New York Times and then just daily tracking around to see what updates they had about how much it’s spreading, how much is it killing people. (1019)
It can seem like it’s information overload if you’re just listening to the news over and over again. But if you’re listening to what they’re saying, the real deal is there hasn’t been a whole bunch. It said social distancing, wash your hands, make sure you you stay at six feet apart. So when you look at the theme of the information it’s actually about a bunch of the same stuff over and over again. (1004)
**Being selective with types of information input**
I try not to read into online social media because a lot of it is misinformation and it really confuses. (1008)
We also have so much information. You have to filter through it and know what’s good and what’s not. I think that also with these articles that get clicked, or liked, or shared, like some of the more sensational or attention-grabbing things, maybe not as much meaningful or well-researched. (1005)
The information has improved since the CDC started putting out ads and bulletins of their own. Our governor also has taken a good approach to try to get good information on a daily basis. You see, the little words from his daily press conferences and check on the numbers, and that’s probably about it for me. (1009)
**Regulating the amount of information**
I don’t watch the news like on overload. I read a couple of articles here and there but I don’t inundate myself to read too much. (1022)
I don’t read all of it. I scan through the headlines and whatever looks interesting or something that I haven’t read before because there’s a lot of duplication. A lot of times, it will be multiple websites just writing about the same thing, so I’m not going to read all of it. (1023)
**Taking breaks**
The amount of information you get and how quickly it changes can be a little overloading. It’s nice to be able to take a break from it. (1001)
In the very beginning, we were watching the news. But I just couldn’t deal with President Trump talking for two hours and then not being able to hear Dr. Fauci and Dr. Burkes. Because they were the ones that I wanted to hear from so I stopped watching it because I couldn’t stand listening to him anymore. (1024)
I don’t watch the news like on overload. I read a couple of articles here and there but I don’t inundate myself to read too much. (1022)

## Data Availability

Deidentified transcripts of the interviews can be obtained from M.A.A. upon request.

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
