# Peer review of "Appraising and Handling COVID-19 Information: A Qualitative Study"

_ijerph, 2021, doi:10.3390/ijerph181910382_

Round 1

Reviewer 1 Report

According to the authors, the paper investigates "how people most impacted by the pandemic 10 have judged and perceived the quality of information regarding COVID-19 and regulated the infor-11 mation flow". This is a too vague topic. Both literature review and research approach do not provide enough information about the domain under study. Results are simple listed without a clear discussion and the implications are complitelly understimated.

Author Response

Thanks for your review.

Reviewer 2 Report

The nature of the research work presented is timely and relevant. It provides an overview of how many people have handled information about the pandemic. However, some aspects should be improved or clarified:

- the title of the article, “Managing COVID-19 Information”, should better reflect the scope of the study. The study highlights some general findings but does not identify the ways in which people dealt with the problems that they faced (how they managed the information);

- the number of people interviewed is quite small, although it constitutes an interesting sample in terms of diversity. However, the reason for not listening to other professionals in the health area or in other areas, such as public transport, hypermarket workers, etc., is not clear. In addition, given the diversity of training, knowledge, and proximity to the pandemic, one would expect a segmentation of opinions that would provide a clearer and more objective identification of each of the interviewed segments, which in some cases corresponds to only one respondent.

- some findings and even conclusions should be better supported, as it is not clear whether they relate to the generality of the opinions given or if they are more evident in some part of the sample;

- the nature of the themes addressed in the questions posed is not perceptible.

Bibliographic references should follow APA standards.

Author Response

Thanks for your review.

Reviewer 3 Report

This is an interesting study. However, I have several comments that would improve this study

  1. The sample size used is very small. 25 people can not provide results that can represent the whole population. I would suggest authors to increase sample size.

  1. The methodology used for analysis is very weak. This study could be better if text mining approached were used. The authors did not even mention about text mining while what they are working on is text data. Several previous studies have used text mining approaches for different purposes. See (Das & Dutta, 2020; Kutela et al., 2021).

  1. Instead of adding the list of all participants, authors could provide the statistics for the study population.

  1. Information in table 2 and table 3 do not add any value to this study.

References

Das, S., & Dutta, A. (2020). Characterizing public emotions and sentiments in COVID-19 environment: A case study of India. Https://Doi.Org/10.1080/10911359.2020.1781015, 1–14. https://doi.org/10.1080/10911359.2020.1781015

Kutela, B., Langa, N., Mwende, S., Kidando, E., Kitali, A. E., & Bansal, P. (2021). A text mining approach to elicit public perception of bike-sharing systems. Travel Behaviour and Society, 24, 113–123. https://doi.org/10.1016/j.tbs.2021.03.002

Author Response

Thanks for your review.
